# Immune Cytolytic Activity and Strategies for Therapeutic Treatment

**DOI:** 10.3390/ijms25073624

**Published:** 2024-03-23

**Authors:** Stephanie Agioti, Apostolos Zaravinos

**Affiliations:** 1Cancer Genetics, Genomics and Systems Biology Laboratory, Basic and Translational Cancer Research Center (BTCRC), 1516 Nicosia, Cyprus; sa231751@students.euc.ac.cy; 2Department of Life Sciences, School of Sciences, European University Cyprus, 1516 Nicosia, Cyprus

**Keywords:** cytolytic activity, tumor-infiltrating lymphocytes, gene expression profiling, spatial analysis techniques, immune checkpoint pathways, tumor microenvironment, antigen presentation, metabolic pathways

## Abstract

Intratumoral immune cytolytic activity (CYT), calculated as the geometric mean of granzyme-A (GZMA) and perforin-1 (PRF1) expression, has emerged as a critical factor in cancer immunotherapy, with significant implications for patient prognosis and treatment outcomes. Immune checkpoint pathways, the composition of the tumor microenvironment (TME), antigen presentation, and metabolic pathways regulate CYT. Here, we describe the various methods with which we can assess CYT. The detection and analysis of tumor-infiltrating lymphocytes (TILs) using flow cytometry or immunohistochemistry provide important information about immune cell populations within the TME. Gene expression profiling and spatial analysis techniques, such as multiplex immunofluorescence and imaging mass cytometry allow the study of CYT in the context of the TME. We discuss the significant clinical implications that CYT has, as its increased levels are associated with positive clinical outcomes and a favorable prognosis. Moreover, CYT can be used as a prognostic biomarker and aid in patient stratification. Altering CYT through the different methods targeting it, offers promising paths for improving treatment responses. Overall, understanding and modulating CYT is critical for improving cancer immunotherapy. Research into CYT and the factors that influence it has the potential to transform cancer treatment and improve patient outcomes.

## 1. Introduction

### 1.1. Background on Cancer and the Immune System

Cancer is the second leading cause of death, worldwide. In 2022, 19,976,499 new cancer cases were recorded with 9,743,832 deaths among both sexes. It is estimated that until 2060, cancer will be the main cause of death, globally [1].

The immune system is a defense mechanism against infected antigens and self-antigens via a suitable balance between the inhibition and activation of immune responses [2]. It also actively participates in cancer prevention, progression, and therapy [3]. Cytotoxic T cells (CTLs) via their T-cell receptors recognize antigens derived from cancer cells and bind to major histocompatibility complex (MHC) molecules on the surface of antigen presenting cells (APCs) [4]. Moreover, the immune system creates immunological memory through adaptive immune responses for a future, more rapidly and effective response against the same antigens [5,6].

Tumors can evade immune response using various mechanisms, including the inhibition of the immune system, inducing T-cell exhaustion, and restricting antigen recognition [7]. Evasion of cancer cells from the immune system can be defined as one of the cancer hallmarks [8]. Furthermore, cancer cells proliferate and grow faster than the immune system can manage, resulting in the escape of an attack from immune cells. To achieve this, cancer cells create in their surroundings a TME, which affects the efficacy of immune cell activation [9].

The TME plays a crucial role in cancer progression and cell migration [10]. It is composed of cancer cells, blood vessels, extracellular matrix (ECM), stromal cells, and immune cells, including natural killer (NK) cells, B cells, T cells, neutrophils, myeloid-derived suppressor cells (MDSC), and tumor-associated macrophages (TAM) [11]. The interplay between the cellular and extracellular components of the TME contributes to the early diagnosis of cancer [12].

Cancer immunotherapy has enabled never-before-seen success rates in durable tumor control and enhanced survival benefits in patients with advanced cancers. There are at least two different forms of immunotherapy: immune checkpoint inhibitors (ICIs) and adoptive cellular therapies (ACTs) [13]. ICIs use specific monoclonal antibodies (mabs), including anti-programmed death-1 (PD-1)/programmed death-ligand 1 (PD-L1) and/or anti-cytotoxic T-lymphocyte antigen-4 (CTLA-4), to treat several cancers. Specifically, anti-PD-1 and/or anti-CTLA-4 have been proved to increase the overall survival (OS) of patients with melanoma [14].

There are currently eight approved ICIs to treat 18 different types of cancer, comprising tumors with a deficient mismatch repair system/microsatellite instability—high (dMMR/MSI-H), lung cancer, renal cell carcinoma (RCC), ovarian cancer, gastric cancer (GC), and Hodgkin lymphoma [15,16,17]. Nevertheless, many of them exhibit resistance to such therapies and do not respond to them.

### 1.2. The Role and Significance of Immune Cytolytic Activity in Tumor Control

CTLs and NK cells can target and kill target cells using various mechanisms, one of which is the granzymes/perforin pathway [18]. More specifically, upon recognition of the targeted cells, CTLs and NK cells release granzymes and perforins to enter them and lead them to apoptosis [19].

Perforins are responsible for opening pores in the membranes of the targeted cancer cells, facilitating the entry of the granzymes into them, which eventually eliminate cancer cells in an apoptotic manner (Figure 1) [20,21]. Cytolytic activity (CYT) is a new index of immune activation within a tumor and it is calculated by the expression levels of GZMA and PRF1 [20,22].

In 2015, CYT levels were quantified and presented by Rooney et al. as a new index of local immune infiltrate across 18 tumor types [23]. The CYT index varies across different cancer types. Roufas et al. quantified the transcript levels of GZMA and PRF1 and showed that these vary tremendously across the different types of cancer; for example, in acute myeloid leukemia, pleural mesothelioma, sarcoma, and stomach cancer have high CYT levels; while on the other hand, in ovarian, liver, thyroid, esophageal, prostate cancers, glioblastoma, glioma, adrenocortical carcinomas, and uveal melanomathese, were very low. Finally, in head and neck cancer, and skin melanoma, CYT levels were considerably higher compared to the equivalent normal tissues [22].

Interestingly, in colorectal cancer (CRC), CYT has been shown associate with different mutational events and -CYT^high^ tumors are related to an increased tumor mutational burden (TMB) [24,25,26] and upregulate the expression levels of immune checkpoints, including PD-1, PD-L1/2, CTLA-4, LAG3, and IDO1 [22,24].

Moreover, -CYT^high^ tumors are associated with a MSI-H status and better survival of patients with metastatic CRC, hepatocellular carcinoma (HCC), and triple-negative breast cancer (TNBC) [24,27,28,29].

CYT has a significant role in cancer control and progression and can also be used as a prognostic biomarker reflecting the immune status in HCC [30], immunity, and clinical outcomes [25]. In addition, CYT^high^ skin melanoma patients have better prognoses [31].

Although available therapies improve the prognosis for many cancer patients, most of them still have a poor prognosis, or develop resistance. Therefore, the responsible mechanisms for the aberrations taking place within the TME need to be investigated further. Herein, we investigated the factors that influence CYT in order to develop better therapeutic approaches against cancer.

Immune CYT plays a pivotal role in cancer patient prognosis and immunotherapy response. The investigation of the mechanisms related to CYT, as well as the assessment of strategies to enhance CYT, will significantly contribute to develop the most effective treatment plan for each patient. Our review highlights the significance of immune CYT in cancer management, summarizing the different methods used to assess immune CYT, the factors influencing it, and its clinical implications.

## 2. Key Concepts in Immune Cytolytic Activity

### 2.1. Methods for Assesing CYT

#### 2.1.1. Tumor-Infiltrating Lymphocytes (TILs)

Isolation and Characterization of TILs

TILs are important components of TME and have the ability to recognize and kill autologous cancer cells, both in vitro and in vivo [32]. Their isolation, especially in the early stages of cancer, is very important [33]. A phenotypic analysis of TILs isolated from RCC showed that TILs were mainly composed of CD4+, CD8+, and CD56+ cells [34]. CYT is strictly related to the population of CD8+ T cells and can be affected upon their activation [35], as well as from the population of TILs within the TME.

In addition, the spatial characterization of TILs has been demonstrated as a key prognostic biomarker to predict treatment response in breast cancer [36]. In addition, in a study by Chew et al. the isolation of TILs from tumor and non-tumor liver tissues of resected HCC patients was used to analyze immune gene expression profiles, and the results show that the activation and proliferation of TILs in an inflammatory immune microenvironment can enhance patient survival and tumor progression [37].

Currently, immunotherapies using TILs have gained attention due to their efficacy and the inefficiency of conventional immunotherapies [38]. As the population of TILs is associated with CYT levels within the TME and both of them are prognostic biomarkers, their combination needs to be investigated.

Analysis of TILs by flow cytometry or immunohistochemistry

TILs can be expanded and analyzed using different in vitro methodologies, including flow cytometry and immunohistochemistry. In a study in 2017, TILs were surgically expanded from 19 patients with pancreatic adenocarcinoma in the presence of IL-2 and were characterized phenotypically by flow cytometry. The results showed that TILs were composed mainly of CD4+ and CD8+ T cells [39].

In 1994, expanded TILs derived from non-small-cell lung cancer (NSCLC) were used in adoptive immunotherapy, following surgery with promising results [32]. Flow cytometric immunophenotypic analysis of TIL immunophenotypes could predict both the clinical effectiveness of immunotherapy [40] and early relapse in localized clear-cell RCC [41].

#### 2.1.2. Gene Expression Profiling

Identification of Cytolytic Markers

GZMA and PRF1 are two key cytolytic effectors, components of the granzyme–perforin pathway, whose transcription levels are used for the measurement of immune CYT, and both of them are significantly affected upon CD8+ T-cell activation [35]. In addition, upon activation, CTLs and NK cells mediate the cytolysis of targeted cells using various mechanisms. One of these is the granzyme–perforin pathway in which CTLs and NK cells, upon their exposure to virally infected/cancer cells, release the afore-mentioned cytolytic toxins, which will open pores and enter the targeted cells leading them to apoptosis [19,42].

Measurement of CYT through mRNA expression

Immune CYT is assessed by the expression levels of cytolytic effectors secreted from CD8+ T or NK cells. It is measured using the mRNA expression levels of GZMA and PRF1 [25,30,43], through quantitative real-time PCR (RT-qPCR) or RNA-sequencing [23]. High levels of the genes encoding for these toxins indicate a robust cytolytic activity, which is critical for the effective elimination of infected or tumor cells.

#### 2.1.3. Spatial Analysis Techniques

Multiplex Immunofluorescence (mIF) and Imaging Mass Cytometry (IMC)

mIF and IMC are tissue imaging techniques used to detect multiple markers on a single tissue slide and their combination has been recently demonstrated as an accurate imaging method for single-cell segmentation [44]. mIF allows the simultaneous accurate detection of a variety of immune markers in different tissue types [45]. It has the unique ability to study the spatial interaction between immune and tumor cells without affecting the architectural features of the tumor [46]. IMC, on the other, is based on the principles of flow cytometry analysis and mass spectrometry and it uses metal isotope-labeled antibodies that enables the highly multiplexed imaging analysis of proteins, isotopes, and markers within tumor tissues [47,48]. In 2021, IMC was used to characterize the TME of metastatic melanoma patients who received immunotherapy in order to find indicative biomarkers of treatment response with very promising results [49].

The interplay between the immune cells and tumor cells is crucial for tumor progression, metastasis, and therapy, and significantly affects the levels of CYT. Both techniques can be used to study how the aberrations taking place within the TME can affect CYT for prognostic or therapeutic purposes.

Among different spatial analysis techniques, both mIF and IMC are very significant in the field of immuno-oncology and are used to study and understand better the TME in order to create more effective therapies and identify predictive biomarkers of response to immunotherapy [50].

Assessment of CYT within the TME

CYT and immune cells within the TME are strictly related. The population of these immune cell types within the TME is strictly associated with better survival in -CYT^high^ tumors [23,30,51]. In addition, in GC, the CYT score was associated with a high infiltration of macrophages and low infiltration of Tregs [25].

Furthermore, CYT^high^ primary and metastatic skin melanomas were enriched in CD8+ T cells, B cells, and M1 macrophages, while CYT^low^ tumors were enriched in CD4+ T cells, monocytes, and NK cells [52]. In glioblastoma, high CYT levels were related to the higher infiltration of immunosuppressive cells, including neutrophils and M2 macrophages, and worse survival [43].

As immune CYT and immune cells are closely associated within the TME, it is considered that immune CYT can be assessed within the TME.

### 2.2. Factors Influencing CYT

#### 2.2.1. Tumor Intrinsic Factors

Tumor Mutational Burden

TMB is defined as the total number of genetic mutations in the DNA of cancer cells [53]. The calculation of TMB is performed by using different methods. Initially, it was measured using whole-exome sequencing and included only non-somatic mutations [54,55]. Apart from the tumor tissue, it can also be measured from blood in cases of circulating tumor DNA (ctDNA). TMB is an important biomarker in oncology because high TMB is related with a better response to immunotherapy, particularly ICIs [53,56]. As tumors with higher mutation rates may present more “foreign” antigens to the immune system, making them more susceptible to immune attack, TMB is a measure of the neoantigen load of a tumor. This, can vary widely across different types of cancer, but also among patients with the same type of cancer. It is worth mentioning that, among different immunotherapies across 27 tumor types or subtypes, highTMB associates with a better response to anti-PD-1 therapy [57].

In 2015, a high TMB was strongly associated with a better response of NSCLC patients to anti-PD-1 immunotherapy and longer progression-free survival (PFS) [58]. Moreover, NSCLC patients with a high TMB score, who received nivolumab (anti-PD-L1), had longer PFS and a higher objective response rate (ORR) than those who received chemotherapy [59]. On top of anti-PD-L1 alone, combination immunotherapy of nivolumab and ipilimumab in high-TMB patients with advanced NSCLC showed a longer PFS than chemotherapy treatment [60].

TMB is also linked with other emerging biomarkers. Specifically, MSI-H/dMMR tumors have been identified to exhibit a high TMB and are also associated with a better response to ICI therapy [61]. Besides MSI-H, a high TMB associated with increased CYT levels and the downregulation of various of immune checkpoints in colon cancer [24].

More and more studies support the idea that TMB is a promising predictive biomarker, and its evaluation plays a key role in immuno-oncology. In addition, TMB can be very valuable in treatment selection for ICI therapy [55].

Antigen Presentation and MHC Expression

The intricate processes of antigen presentation and MHC (major histocompatibility complex) expression are foundational to the immune system’s ability to recognize and respond to cancer cells. Specifically, the presentation of tumor antigens via MHC class I molecules to CTLs is a critical step in the initiation of CYT. This interaction is essential for CTLs’ recognition of cancer cells as targets for cytolysis.

CYT is a potent indicator of the immune system’s capacity to mount an effective response against tumor cells, the efficacy of which is heavily modulated by the efficiency of antigen presentation and the expression levels of MHC molecules on tumor cells. Tumors may evade immune surveillance by downregulating the expression of their MHC class Imolecules, thereby impairing the immune system’s ability to detect and eliminate cancer cells. This evasion strategy highlights the importance of the MHC in the regulation of CYT and underscores the potential therapeutic value of enhancing MHC expression to boost CYT and improve patient outcomes.

Furthermore, the role of professional APCs in priming and activating CTLs cannot be overstated. By presenting cancer antigens in the context of MHC class I and II molecules, APCs initiate the cytolytic process that targets tumor cells for destruction [62,63]. This highlights the interdependence of antigen presentation, MHC expression, and CYT in the orchestration of an effective anticancer immune response.

In therapeutic contexts, strategies to enhance antigen presentation and MHC expression have shown promise in boosting CYT and overcoming immune evasion by tumors. For example, treatments that increase the visibility of tumor antigens to the immune system, thereby facilitating their recognition and destruction by CTLs, directly contribute to elevating CYT levels within the tumor microenvironment (TME) [64]. This approach not only underscores the therapeutic potential of targeting antigen presentation pathways, but also highlights the crucial role of the MHC in modulating immune responses against cancer.

#### 2.2.2. Tumor Microenvironment Factors

Immunomodulatory Cells in the TME and CYT

The TME is a complex network of cellular and molecular interactions that profoundly influences immune surveillance and cancer progression. Within the TME, a myriad of factors interact to modulate the CYT of immune cells. This cytolytic activity plays a pivotal role in the immune-mediated destruction of tumor cells. One of the critical components influencing CYT within the TME, is the presence of immunosuppressive cells, such as regulatory T cells (Tregs) and myeloid-derived suppressor cells (MDSCs). These cells can inhibit the functions of CTL and NK cells, thus dampening CYT. This is performed through various mechanisms, including the secretion of inhibitory cytokines and the expression of immune checkpoint molecules [65].

Targeting these immunosuppressive pathways has emerged as a promising strategy to enhance CYT. For instance, the depletion of Tregs or the inhibition of MDSC functions have been shown to augment antitumor immunity by enhancing CYT [66,67].

Furthermore, the expression of immune checkpoint molecules, such as PD-1 on CTLs and its ligand PD-L1 on tumor cells, represents another layer of regulation of CYT within the TME. The binding of PD-1 to PD-L1 inhibits the activity of CTLs, leading to reduced CYT and allowing tumor cells to evade immune destruction. The blockade of the PD-1/PD-L1 axis with monoclonal antibodies has been a groundbreaking approach to cancer therapy, leading to the reinvigoration of CYT and significant clinical benefits across various cancer types [68,69].

The architecture of the TME, including the extracellular matrix (ECM) and stromal cells, also plays a critical role in modulating CYT. The ECM can act as a physical barrier to immune cell infiltration, while stromal cells can secrete factors that either support or inhibit immune cell recruitment and function. Modifying the ECM or targeting stromal signals has been proposed to enhance CYT by improving immune cell access to tumor cells and fostering a more immunogenic TME [70,71].

Lastly, the role of cancer stem cells (CSCs) in regulating CYT is an area of growing interest. CSCs are thought to employ various mechanisms to evade immune detection and destruction, including the modulation of antigen presentation and the induction of immunosuppressive microenvironments. Understanding how CYT can be harnessed to target CSCs is critical for developing therapies that prevent tumor recurrence and metastasis [72,73].

Immune Checkpoints and CYT

Immune checkpoint proteins are expressed on the surface of cancer cells to inhibit T-cell-mediated immune responses and are currently used to treat different cancer types [74]. ICI therapy is a very promising type of therapy that employs mabs against these immune checkpoints to activate T cells, allowing them to kill cancer cells [75]. There are currently different immune checkpoint inhibitors used to treat ~15 different types of tumors [76]. The most well-studied ICIs are anti-CTLA-4 (Ipilimumab), anti-PD-1 (Nivolumab, Pembrolizumab, Cemiplimab), and anti-PD-L1 (Atezolizumab, Durvalumab) mabs [75,77] (Figure 2). However, there is currently an increasing diversity in treatment options, with new ICIs that enhance cancer treatment efficacy across a broad range of tumor types, such as anti-LAG-3 (Relatlimab) [78], anti-TIGIT (Tiragolumab) [79] and anti-TIM-3 mabs [80].

It has been reported that CYT is positively correlated with the expression of immune checkpoints and it can be used as a predictive biomarker for response to ICI therapies. Specifically, CYT^high^ CRC, EC, and NSCLC tumors have increased expression of immune checkpoints, making them better candidates for response to ICI therapies, compared to CYT^low^ ones [26,27,81].

Moreover, the downregulation of the expression of immune checkpoints directly affects CYT and it can be used as a mechanism of immune evasion by tumors. It is indicative that CYT^low^ melanomas downregulate the expression of several immune checkpoints, including CTLA-4 and PD-1 [52].

Stromal and ECM Components Influencing CYT

Stromal cells have fundamental roles in health and disease and their interaction with cancer cells are crucial for cancer initiation (Figure 3) [82]. The interaction between ECM components is crucial in different cellular processes and functions, including proliferation, differentiation, migration, and survival [83].

In addition, the interplay between stromal cells and other ECM components can either enhance or reduce cytolytic response against tumor cells. In carcinomas, stromal cells alter ECM formation resulting to tumor progression. Moreover, the crosstalk of stromal cells and immune cells within the TME enhance immune evasion and reduce antitumor activity [84].

Mesenchymal stromal cells, a subpopulation of stromal cells, remodel ECM components within the TME and regulate immune responses [85]. In gliomas, immune CYT is a potential biomarker for the mesenchymal subtype [86].

Another form of stromal cell that is present in the TME is cancer-associated fibroblasts (CAFs), which promote tumor growth, invasion, angiogenesis, and metastasis through the production of VEGFA, PDGFC, FGF2, CXCL12, osteopontin, and CSF3 [87]. Thus, CAFs within the TME can modify the ECM and constitute a protective mechanism of cancer cells from infiltrating T cells. Furthermore, their presence reduces the number of CTLs and their cytolytic activity within the TME [88].

Cancer Stem Cells (CSCs) and CYT

CSCs constitute a sub-group of cells within tumors with the abilities of differentiation, self-renewal, and tumor progression [89] (Figure 3). Leucine-rich repeat-containing G protein-coupled receptor 5 (Lgr5)+ stem cells, a sub-population of CSCs, are highly expressed in different types of cancer; their proliferation is associated with cancer occurrence, progression, and metastasis, and their regulation can be used for therapeutic purposes [90].

The crosstalk of CSCs with infiltrating immune cells within the TME enhances tumor progression and immune evasion [91]. It has been experimentally reported that the presence of CSCs can increase the cytolytic activity of NK cells in malignant melanoma cell lines, indicating that the targeting of CSCs can be used to enhance NK-based adoptive immunotherapies in metastatic melanoma patients [92].

Lgr5+ constitutes a marker for CSCs within the TME and their monitoring can provide a new treatment option for liver cancer patients by targeting the PTEN/AKT and Wnt/β-catenin signaling pathways. Specifically, in liver cancer, studies have shown that the activation of the above-mentioned signaling pathways increases the proliferation of Lgr5+ cells, while their inhibition decreases their population [93,94].

As Lgr5+ cells are overexpressed in various cancer types and constitute an important component of the TME, they are probably related to immune CYT. The interplay between CYT and Lgr5+ cells within the TME can present a new chapter in cancer management. In addition, the investigation of the relationship between immune CYT and Lgr5+ cells, as well as the exploration of the TME to see how the increase and/or decrease in the levels of immune cytolytic activty affect the population of Lgr5+ cells, is crucial in metastasis and cancer treatment.

#### 2.2.3. Host Factors

Host factors play a pivotal role in modulating the immune system’s effectiveness in recognizing and eliminating tumor cells, directly influencing CYT. Genetic variations, immune system composition, and overall health can significantly impact the ability of CD8+ T cells and NK cells to execute their cytolytic functions efficiently.

As regards genetic predisposition, single nucleotide polymorphisms (SNPs) within genes encoding for immune regulatory elements can alter the immune response against tumors by affecting the proliferation, activation, and cytolytic efficiency of CD8+ T cells. These genetic variations can therefore dictate the magnitude of CYT against cancer cells, underscoring the importance of personalized medicine approaches in cancer immunotherapy [23,68].

Moreover, the overall composition and health of the immune system, determined by both intrinsic (genetic) and extrinsic (environmental) factors, significantly affect CYT. For example, the presence and activity of CD8+ T cells within the TME are crucial for an effective antitumor cytolytic activity. Studies have shown that the density and functionality of TILs, especially CD8+ T cells, are positively correlated with CYT and patient prognosis in various cancers [95,96]. These observations highlight the importance of host factors in shaping the immune landscape and the effectiveness of CYT against cancer.

Furthermore, at the molecular level, the engagement of CD8+ T cells with tumor cells through antigen presentation and MHC-I interaction is essential for the induction of CYT. The process by which CD8+ T cells recognize tumor antigens presented by MHC-I and subsequently execute their cytolytic function is fundamental to the immune-mediated elimination of cancer cells. The efficiency of this process can be influenced by host genetic factors, such as MHC polymorphisms, which can alter antigen presentation efficacy and subsequently CYT [97].

In light of these considerations, targeting host factors to enhance CYT represents a promising avenue for cancer therapy. Strategies aiming to boost the functionality and presence of CD8+ T cells in the TME, such as ICI, adoptive cell transfer (ADT), and vaccination, are based on understanding and manipulating host factors to maximize CYT [98,99].

### 2.3. Clinical Implications and Prognostic Significance

#### 2.3.1. Association between CYT and Patient Outcomes

CYT is closely related with patient prognosis and outcomes and the expressions of both GZMA and PRF1 genes synergistically, or alone can affect the OS of cancer patients [24,100,101,102] (Table 1).

Recently, in breast cancer, it was reported that the overexpression of GZMA alone increased the infiltration of dendritic cells (DCs) and CD8+ T cells and was correlated with better OS, DSS, and progress-free interval (PFI) [104]. In addition, Roufas et al. reported that CYT^high^ in skin melanomas had activated immune-related genes and increased levels of CD8+ T cells, NK cells, B cells, M1 macrophages, and DCs [22].

CYT was also evaluated between tumor and tumor-free tongue patients with squamous cell carcinomas of the oral tongue, and it is remarkable that tumor-free tongue patients had higher CYT levels with better survival, contralateral to the tumor patients in which CYT was not predictive for their survival [51].

#### 2.3.2. Predictive Value of CYT for Immunotherapy Response

In recent years, cancer immunotherapy has enabled never-before-seen success rates in durable tumor control and enhanced survival benefits in patients with advanced cancers. Importantly, ICI therapies using mabs to treat cancer patients has created overwhelming enthusiasm [105,106].

CYT is identified as a new biomarker of immunotherapy reflecting an antitumor immune response [23] and it can be used to predict the patient response to ICI therapies. It is indicative that the expression levels of both CYT markers (GZMA and PRF1) can predict the favorable prognosis of cancer patients who receive ICI therapy [107].

High-CYT levels within CRC, EC, and NSCLC have been associated with a high expression of immune checkpoints, including PD-L1 and CTLA-4, and these patients could respond better to such therapies [26,27,81]. It was also reported that CYT^high^ GC tumors responded better to anti-PD-1 therapy [25], and CYT^high^ CRC patients were more sensitive to ICI therapies compared to CYT^low^ ones [24].

In addition, CYT^high^ skin melanoma patients who received anti-CTLA-4 and/or anti-PD-1 therapy had better clinical results due to a higher immunophenoscore, compared to those with low CYT levels [52]. Interestingly, metastatic skin melanoma patients with high GZMA expression levels exhibited significantly higher levels of CTLA-4, PD-L1, and PD-L2, and had a better response to anti-PD-L1 (Nivolumab) treatment, resulting in better clinical benefits and long-term survival [108,109].

Furthermore, in skin melanoma patients being treated with anti-CTLA4 mab (Ipilimumab), it was shown that the intratumoral immune cytolytic levels increased the infiltration of CD8+ T cells and the expression of MHC class I molecules [110]. In addition, high GZMA expression was shown to lead to immune activation and the infiltration of CD8+ T cells, rendering patients more sensitive to anti-PD-L1 immunotherapy [111].

Furthermore, CYT^high^ prostate cancer patients are more sensitive to ICIs due to the fact that high GZMA and PRF1 expression levels increase the number of CD8+ T cells and the expression of immune checkpoints, including PD-L1, compared to patients with low CYT levels [112].

Although, the different forms of immunotherapy have achieved success in treating patients with different cancer types, only a small number of them responds to these therapies and some create resistance. Therefore, the evaluation of CYT as a prognostic factor in treatment selection and immunotherapy response will help cancer patients to receive the most appropriate therapy for their cancer, with the best results.

#### 2.3.3. Integration of CYT Assessment into the Clinical Practice

Potential Biomarkers for CYT Evaluation

Apart from the expressions of GZMA and PRF1, CYT is also significantly related to the expression of other different biomarkers. Specifically, the TME plays a crucial role in cancer progression and its composition can predict a patient’s prognosis. Therefore, CYT depends on the proportion of immune cells within the TME, which can be used as a biomarker for CYT evaluation. Notably, it has been recorded that the presence of T cells within the TME relates to a favorable prognosis [113], and CYT is positively correlated with the proportion of tumor-infiltrating CD8+ T cells and M1 macrophages.

Additionally, high CYT levels are strongly related to a high infiltration of the above-mentioned immune cells [22,29,110,111,112,114]. Apart from GZMA and PRF1, GZMB is also highly expressed in CD8+ T cells [115,116].

In addition, CYT is positively associated with the status of MSI TMB and the rate of tumor-mutated peptides, called neoepitopes. Rooney et al. first showed that cancer neoepitopes were associated with CYT in a number of tumors [23]. CYT was also later shown to correlate with an increased mutational burden and neoepitope load in gastric and colorectal tumors [24,25].

Moreover, MSI-H colorectal tumors are related to increased levels of both CYT and mutations [24] and dMMR/MSI-H colorectal adenocarcinoma patients were shown to respond better to anti-PD-1 immunotherapy [117]. In contrast, in pancreatic ductal adenocarcinoma, the increase in CYT levels does not correlate with an increase in TMB or neoepitope load [20].

ICIs against CTLA-4 and PD-L1 have been proposed to be important indicators for both CYT^low^ and CYT^high^ cancer patients. Most studies support that high levels of PD-L1 and CTLA-4 are present in CYT^high^ cancer patients [26,27,81], being more sensitive to these therapies [24].

Role of CYT in Patient Stratification and Treatment Selection

The standard treatment option for cancer therapy includes immunotherapy, chemotherapy, radiotherapy, surgical removal, and/or targeted therapy, depending on the tumor’s stage [118]. CYT plays a pivotal role in patient stratification and its levels have a significant value to choose the suitable treatment for each patient.

As mentioned above, CYT^high^ patients have different responses to immunotherapies compared to CYT^low^ ones, as high CYT levels increase the activation and infiltration of CD8+ T cells [22,110,111,112,114].

CYT is significantly related to the amount of active CTLs that can be used in treatment selection. In vitro studies of Paclitaxel, Doxorubicin, and Cisplatin (CIS) showed that all of the above-mentioned drugs enhance cell-mediated killing by CTLs [119,120,121]. It has also been reported that low non-cytotoxic doses of 5-Fluorouracil, Taxotere, Cisplatin, Irinotecan, and CIS induce antitumor responses of CD8+ T cells, which are connected to high cytolytic levels in colon cancer [122]. On the contrary, CYT^low^ androgen receptor (AR) and estrogen receptor (ER)-positive breast cancer patients were associated with low infiltration and CD8+ T cells and lower overall anticancer immunity and survival [123].

In many patients with different types of cancer, the conventional therapies do not provide satisfactory clinical results. Therefore, in recent years, the combinational therapy of cancer immunotherapy and chemotherapy has gained credibility due to its high efficacy, especially in the treatment of patients with advanced stages of cancer. CYT can be used to predict the response of cancer patients to ICI therapies. Specifically, CYT^high^ cancer patients exhibited significantly higher levels ofPD-L1 and CTLA-4, among other immune checkpoints, and responded better to ICI therapies [25,26,27,52,81,108,109,111,112].

Furthermore, anti-CTLA-4 therapy alone and/or with anti-PD-1 treatment has been shown to offer important therapeutic benefits to cancer patients [124], and it enhances antitumor immune activity due to an increase in the number of CD8+ and CD4+ TILs. Moreover, anti-CTLA-4 therapy combined with Cisplatin chemotherapy was significantly more effective. In addition, it increased the expression of perforins and granzyme B and improved the survival of patients, compared to anti-CTLA-4 and Cisplatin therapies in murine mesothelioma [125].

Even though there is an improvement in the prognosis for many patients depending on the stage of cancer, most cancer patients still have a poor prognosis, or develop resistance to immunotherapy. CYT is a relatively new index that can be used as a valuable tool in cancer patient treatment plans.

### 2.4. Modulation of CYT

#### 2.4.1. Current and Emerging Immunotherapy Strategies

Immune Checkpoint Inhibitors (ICIs)

ICIs have made tremendous progress since their approval and are presently used in cancer immunotherapy [126,127].

Ipilimumab (anti-CTLA-4) constitutes the first checkpoint inhibitor that was approved for cancer treatment, significantly improving the OS of metastatic skin melanoma patients [128,129]. Moreover, the use of Pidilizumab (anti-PD-1) showed a 1-year better OS in metastatic melanoma patients who were previously treated with Ipilimumab [130].

Adoptive Cell Therapies (ACTs)

ACT refers to a type of personalized cancer immunotherapy in which lymphocytes are expanded and grown exvivo and re-infused into cancer patients [131,132], providing a possibly dynamic general treatment [133].

ACT using TILs or gene-modified T cells expressing T-cell receptors or chimeric antigen receptors (CAR T-cells) is an alternative type of therapy that triggers the immune system to recognize and kill cancer cells, and it has shown promising results in many cancer types, particularly in hematological malignancies [134].

For example, in leukemia, the use of CAR T-cell therapy targeting CD19, a protein found on the surface of B cells, has led to durable responses in chronic lymphocytic leukemia (CLL) and B-cell acute lymphoblastic leukemia (B-ALL). In B-ALL, clinical trials have shown remission rates as high as 83% in patients with a median follow-up of 29 months [135]. The FDA’s first approval of CAR T-cell therapy was for pediatric ALL, following the remarkable success story of Emily Whitehead, the first pediatric patient to receive CAR T-cell therapy. This pioneering treatment has since led to the exploration of CAR T-cell therapy in other pediatric cancers, including acute myeloid leukemia (AML) and neuroblastoma.

CAR T-cell therapy has also achieved remarkable success in treating various types of lymphoma, including diffuse large B-cell lymphoma (DLBCL). The therapy has become a frequently used treatment for several lymphoma subtypes due to its effectiveness [136]. In addition, the FDA has approved CAR T-cell therapies targeting B-cell maturation antigen (BCMA) for the treatment of multiple myeloma, based on promising preclinical and clinical data.

Despite these successes, the application of CAR T-cell therapy to solid tumors has faced more challenges due to issues like the TME, the identification of suitable target antigens, and tumor heterogeneity. However, ongoing research and clinical trials are focused on overcoming these obstacles and expanding the potential uses of CAR T-cell therapy [135].

Vaccines and Oncolytic Viruses

Oncolytic viruses (OVs) constitute a promising category of cancer therapy due to their ability to induce immunogenic cell death (ICD) and to activate tumor-specific T-cell responses, leading to the eradication of cancer cells [137,138]. OVs also modulate the TME by increasing the maturation of tumor-specific T cells within it due to the high number innate immune cells [139].

OVs can be used as immunotherapeutic anticancer vaccines to enhance tumor-specific T-cell responses and mediate the killing of unharmed cancer cells [140]. Moreover, it has been shown through both in vitro, in vivo, and preclinical studies that the combination of OVs with anti-CTLA-4 and/or anti-PD-1 ICIs has high oncolytic efficacy [137,141].

The death of cancer cells induced by OVs can lead to the release of damage-associated molecular patterns (DAMPs), which are crucial for eliciting antitumor immune responses. For example, an oncolytic virus engineered to target telomerase induced autophagic cell death in tumor cells, which was found to be immunogenic. Such mechanisms contribute to the activation of the immune system against cancer, underscoring the therapeutic potential of OVs in cancer treatment [139].

A study demonstrated that preconditioning with low-dose Cyclophosphamide before administering viral cancer vaccines could significantly enhance tumor-specific CD8+ T-cell responses. This approach led to a notable reduction in metastatic burden and improved survival in a murine model of disseminated pulmonary melanoma, showcasing how Cyclophosphamide can potentiate specific cellular immunity and the efficacy of OV-based vaccines [142].

Another innovative approach involved an oncolytic virus engineered to express PD-L1 inhibitors, which could bind to PD-L1+ tumor cells and immune cells in both autocrine and paracrine manners. This strategy facilitated the activation of tumor neoantigen-specific T-cell responses, highlighting the potential of armed OVs to overcome immune evasion mechanisms employed by tumors and to enhance the efficacy of immunotherapeutic interventions [143].

#### 2.4.2. Combination Approaches to Enhance CYT

Dual Immune Checkpoint Inhibition (ICI)

Despite the fact that ICI therapies have received increased interest, their efficacy alone is restricted by low response rates. Dual ICI constitutes one of the most promising forms of immunotherapy for cancer treatment that enhances the antitumor activity of cytotoxic T cells, and it is already used to treat solid tumors with great efficacy and promising results, including skin melanoma and RCC, as well as advanced HCC [144].

Different combinatorial therapies with mabs are used to treat cancer both in preclinical models and clinical trials. The combination of anti-CTLA-4 and anti-PD-1 mabs can enhance antitumor activity by increasing the infiltration of CD8+ T cells and decreasing the number of Tregs. As a result of this, the TME converts from being immunosuppressive to inflammatory with better clinical results [145,146]. Moreover, in NSCLC patients, immunotherapy using Nivolumab (anti-PD-1) and Ipilimumab (anti-CTLA-4) mabs showed survival benefits [147].

On the contrary, in head and neck squamous cell carcinomas, the combinatorial immunotherapy with anti-PD-1/CTLA-4 mabs was shown to cause a decrease in the activation of CD8+ T cells and Tregs [148]. In addition, phase II clinical trials of concurrent treatment with the above-mentioned mabs showed higher response rates and PFS, compared to the monotherapies in melanoma patients [149,150].

Therefore, dual ICI is a powerful therapy and it can be considered almost certain that, in the near future, its use will increase with better success rates.

Modulation of the TME

The TME is involved in different biological processes implicated with the initiation and treatment and it constitutes an important tool for both the prevention and treatment of cancer [151,152]. The modulation of the TME is crucial in different stages of cancer, including the growth of a primary tumor, the development of metastasis, and immune evasion, and it plays a leading role in strategies for cancer treatment [153].

The TME is modulated by cancer cells due to their high consumption of glucose. This metabolic modulation of the TME increases the expression of immune checkpoints and the infiltration of TILs [154]. Moreover, the treatment of cancer through the immune modulation of the TME makes headway, and studies focus on this to understand more about the TME components and their interactions between themselves in order to use it as a prognostic tool. The TME is already a prognostic tool for enhancing cancer immunotherapy and it used to predict its efficacy [155]. For example, the research indicates that modulating the TME, through approaches like targeting adenosine pathways or adjusting CD8+ T-cell metabolism within the TME, can significantly increase the success of immunotherapeutic treatments [156].

Furthermore, the composition and characteristics of the TME can serve as prognostic tools for predicting the efficacy of immunotherapy. A systematic evaluation and construction of machine learning models based on the tumor microenvironment have been used to predict prognosis and immunotherapy efficacy in triple-negative breast cancer, showcasing the potential of leveraging TME insights for clinical decision making [157].

Additionally, lactic acid metabolism within the TME has been identified as a significant factor influencing the efficacy of immunotherapy and the overall prognosis of lung adenocarcinoma patients. A lactic acid metabolism-related gene signature has been found to contribute to predicting immunotherapy outcomes and understanding the TME, underscoring the metabolic aspects of the TME in cancer therapy [158].

These findings emphasize the complexity of the TME and its impact on cancer treatment outcomes. By further understanding and manipulating the TME, it may be possible to enhance the efficacy of immunotherapies and develop more precise prognostic tools for cancer patients.

#### 2.4.3. Challenges and Future Directions of CYT Modulation

CYT can be used to predict the response of cancer patients to ICI immunotherapies, as well as measure immune activation within the tumor. Therefore, its use is expected to achieve an understanding of non-response or resistance to immunotherapies that occurs in different cancer patients.

Nevertheless, modulating CYT presents significant challenges and opportunities for advancing cancer immunotherapy. A key hurdle is the complex interplay between cytolytic immune response and an immunosuppressive TME. Studies have shown that while a high CYT correlates with potential immunotherapeutic efficacy, it is also associated with an immunosuppressive TME and reduced survival in conditions like glioblastoma, underscoring the dual nature of immune responses within the TME [43].

Furthermore, the efficacy of OVs is limited by their ability to induce immunogenic cell death forms, like necroptosis and pyroptosis, efficiently. Although strategies to enhance these cell death pathways show promise, they also highlight the need for a detailed understanding and manipulation of OVs to fully harness their potential [159].

Future directions include the development of engineered biomaterials for targeted immune modulation, which could address the delivery and efficacy issues associated with current therapies. Moreover, understanding and targeting specific immune cell states, such as ‘stem-like’ CD8+ T cells, can lead to more effective and targeted immunotherapies [160]. As the research progresses, a deeper understanding of the mechanisms underlying CYT and the TME’s role will be crucial for overcoming the current challenges and enhancing the effectiveness of cancer immunotherapies.

## 3. Conclusions

Here, we summarize the role of intratumoral immune CYT in cancer progression and its implication in cancer patient prognosis and treatment outcomes. CYT reflects a tumor’s immune activation and plays a significant role in its control. Different methods are used to evaluate CYT, including the isolation, characterization, and analysis of TILs by flow cytometry and gene expression profiling. In addition, CYT can be calculated within the TME by mIF and IMC, and the immune cells that compose the TME constitute an indicator for assessing the CYT.

Importantly, CYT is influenced by different factors, such as the TMB, where cancer patients with high TMBs (especially MSI-H/dMMR ones) show better responses to ICI therapies. Finally, CYT levels are strictly associated with antigen presentation and MHC I and II complexes, as well as TME factors, including immunosuppressive cells, immune checkpoints, stomal cells, and ECM components.

CYT is expected to play a central role in cancer treatment plans in the near future due to its high clinical value. Currently, it is a key prognostic biomarker for patient response in cancer therapies. In different types of cancer, high cytolytic levels associated with better survival [15,22,26,27,28,29,51,103,161,162]. On the contrary, only CYT^high^ glioblastoma patients were related with poor prognosis and worse survival [43,163].

Moreover, CYT is an accurate immunotherapy biomarker that is strictly related to the expression of immune checkpoints. Importantly, CYT^high^ tumors are expected to exhibit a significantly higher response to ICIs. It has been indicated that high expression levels of GZMA and PRF1, synergistically or alone, can increase the levels of CTLA-4 and/or PD-1/PD-L1, resulting in better clinical outcomes and survival for CYT^high^ patients who receive anti-CTLA-4 and/or anti-PD-1/PD-L1 immunotherapies (Figure 4).

Most studies show that CYT^high^ cancer patients have better clinical outcomes. In the near future, more research into CYT is needed, which will focus on recapitulating its importance in cancer immunotherapy, either with dual ICB or by targeting its metabolic pathways to promote an immune-stimulatory microenvironment.

Considering all of the factors above, the current review highlights the significance of understanding the implication of intratumoral immune CYT in cancer in order to use it for therapeutic purposes.

## Figures and Tables

**Figure 1 ijms-25-03624-f001:**
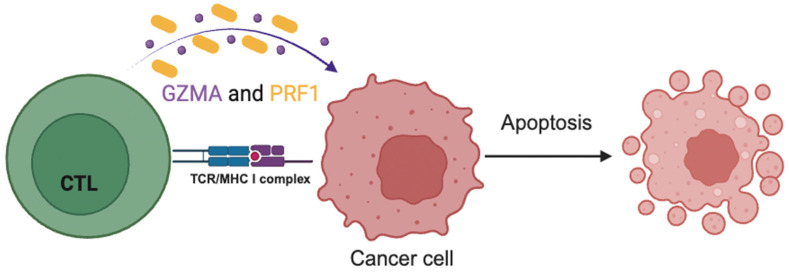
CTLs release granzymes (GZMAs) and perforin (PRF1) against cancer cells upon recognizing them, leading to their apoptosis.

**Figure 2 ijms-25-03624-f002:**
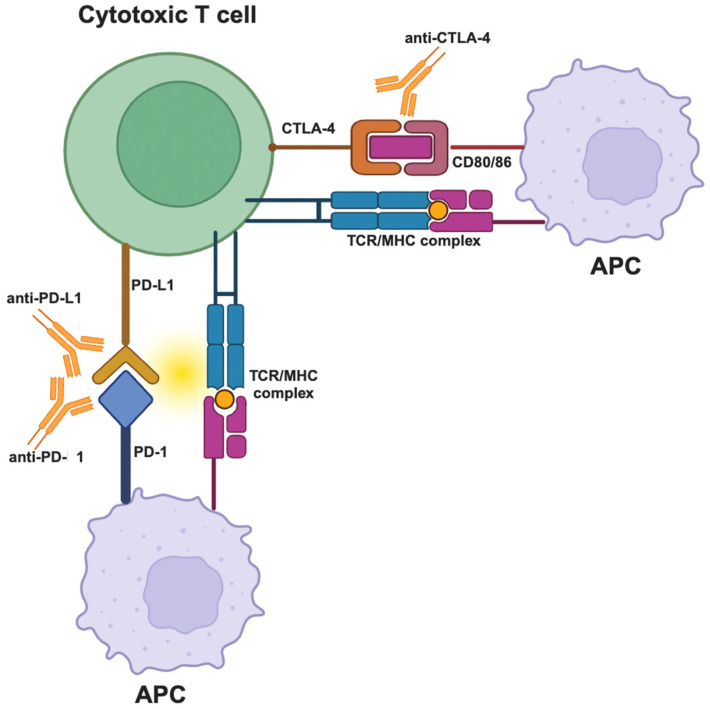
Blockade of inhibitory signals: by blocking the PD-1/PD-L1 axis or the CTLA-4/CD80/86 axis. The antibodies disrupt the inhibitory signaling pathways, which normally suppress the activity of T cells and allow cancer cells to evade immune detection.

**Figure 3 ijms-25-03624-f003:**
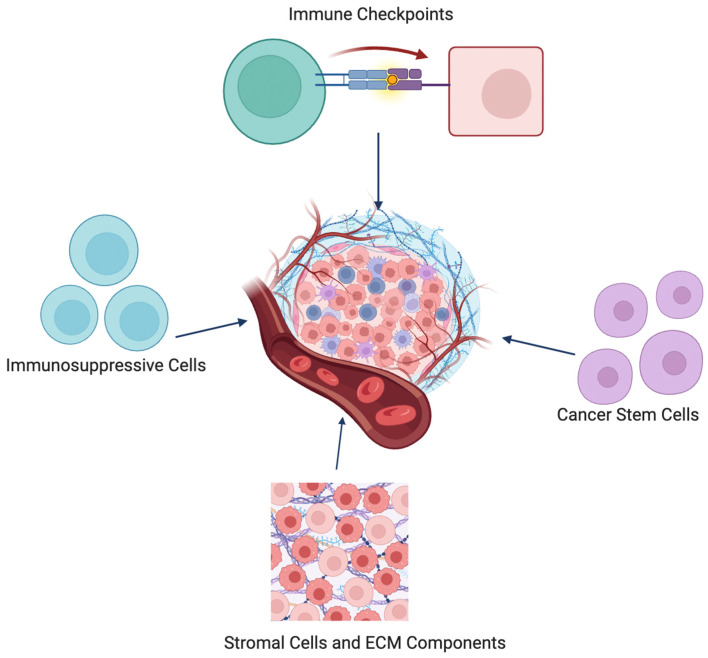
Tumor microenvironment factors influencing CYT.

**Figure 4 ijms-25-03624-f004:**
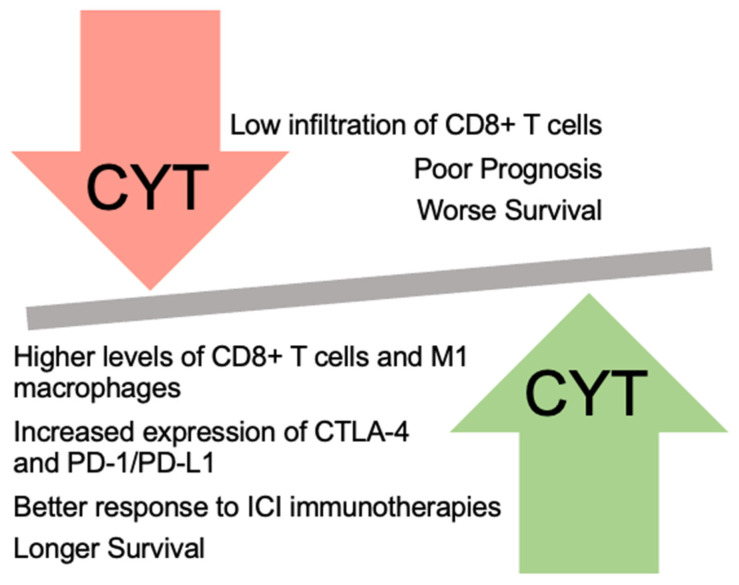
CYT^high^ and CYT^low^ levels affect the infiltration of immune cells, patients’ prognosis, and survival.

**Table 1 ijms-25-03624-t001:** Studies on the association between CYT and patient outcomes.

Reference	Type of Cancer	Association between CYT and Patient Outcomes
[22]	Adrenocortical Carcinoma, Skin Melanoma, and Bladder Cancer	High CYT associated with better patient outcome and OS
[26]	Endometrial Carcinoma	High levels of GZMA and PRF1 synergistically related with better OS
[28]	Triple-Negative Breast Cancer	High CYT significantly related to better OS and disease-specific survival (DSS)
[29]	Hepatocellular Carcinoma	CYT^high^ hepatocellular carcinoma demonstrates significantly longer progress-free interval (PFI), DFI, DSS, and OS
[30]	Hepatocellular Carcinoma	CYT^low^ patients associated with significantly shorter 5-year recurrence-free survival compared to CYT^high^ patients
[43]	Glioblastoma	CYT^high^ patients have worse OS compared to CYT^low^ patients
[103]	Breast Cancer	Overexpression of GZMA related to better OS, DSS, and PFI

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
