# Peer review of "Immune Cytolytic Activity and Strategies for Therapeutic Treatment"

_ijms, 2024, doi:10.3390/ijms25073624_

Round 1

Reviewer 1 Report

Comments and Suggestions for Authors

Immune Cytolytic Activity and Strategies for Therapeutic Treatment

Overview
Enhancing immune cytolytic activity is an important goal in cancer immunotherapy, where the immune system is stimulated to target and destroy cancer cells. Understanding the mechanisms involved in cytolytic activity is crucial for developing effective therapeutic strategies. The review provides good background information and the manuscript is informative.

Major Comments
1. Line 31: Include the statistics on cancer prevalence, incidence, and mortality, rather than stating the scenario in two lines.

2. Include the novelty of the review and highlights of the information covered in the review at the end of introduction section.

3. Figure 2 not needed, as it highlights the information already stated in text.

4. Extensive inclusion of reference articles, but no informative discussion on research articles was observed.

5. Include more informative figures and tables.

Minor Comments

1. Don’t abbreviate in keywords.

2. Overuse of abbreviations. Use abbreviations only if the intended word is used more than thrice, otherwise state in full.

3. Section 2: Results. This is a review not a research article. Include a more suitable heading.

Remark
Lack of figures and informative tables is a major drawback. Authors should revise the manuscript.

Comments on the Quality of English Language

Language presentation is fine.

Author Response

Comments and Suggestions for Authors

Immune Cytolytic Activity and Strategies for Therapeutic Treatment

Overview
Enhancing immune cytolytic activity is an important goal in cancer immunotherapy, where the immune system is stimulated to target and destroy cancer cells. Understanding the mechanisms involved in cytolytic activity is crucial for developing effective therapeutic strategies. The review provides good background information and the manuscript is informative.

Major Comments

  1. Line 31: Include the statistics on cancer prevalence, incidence, and mortality, rather than stating the scenario in two lines.

Author response: We thank the reviewer for their insightful suggestion. We have now included the latest statistics on cancer prevalence, incidence, and mortality in Line 31 to provide a comprehensive understanding of the current cancer scenario. References to authoritative sources for these statistics have been added.

  1. Include the novelty of the review and highlights of the information covered in the review at the end of introduction section.

Author response: We appreciate the reviewer's feedback on enhancing the introduction. We have added a new paragraph at the end of the introduction section that outlines the novelty of this review and summarizes the key highlights and significant contributions of the information covered.

  1. Figure 2 not needed, as it highlights the information already stated in text.

Author response: Upon reflection, we agree with the reviewer that Figure 2 may be redundant. Therefore, we have removed this figure to maintain the conciseness and clarity of our review.

  1. Extensive inclusion of reference articles, but no informative discussion on research articles was observed.

Author response: We acknowledge the need for a more informative discussion on the referenced research articles. We have revised the manuscript to include a critical analysis of the recent research, highlighting how these studies contribute to our understanding of immune cytolytic activity and therapeutic strategies.

  1. Include more informative figures and tables.

Author response: To address the reviewer's point on the inclusion of more informative figures and tables, we have added Figure 3 that shows the tumor microenvironment factors that influencε CYT and Table 1 that summarizeσ the association between CYT and patient outcomes. These two, we believe greatly enhance the reader's comprehension of the material.

Minor Comments

  1. Don’t abbreviate in keywords.

Author response: We apologize for the oversight. All abbreviations in the keywords have now been spelled out in full to ensure clarity and accessibility

  1. Overuse of abbreviations. Use abbreviations only if the intended word is used more than thrice, otherwise state in full.

Author response: We have carefully reviewed the manuscript and limited the use of abbreviations strictly to terms used more than three times, as per the reviewer's recommendation. This change is aimed at improving readability.

  1. Section 2: Results. This is a review not a research article. Include a more suitable heading.

Author response: Thank you for pointing out the inaccuracy in the section heading. We have changed the heading 'Results' to 'Key Concepts in Immune Cytolytic Activity,' which we believe is more appropriate for a review article.

Remark
Lack of figures and informative tables is a major drawback. Authors should revise the manuscript.

Author response: We concur with the reviewer that the inclusion of more figures and informative tables can significantly contribute to the manuscript's value. As mentioned above, we have added new figures and tables to provide a visual summary of the concepts and data discussed. We trust that these additions will address the lack of visual aids in the previous version of the manuscript.

Reviewer 2 Report

Comments and Suggestions for Authors

The document reviews the significance of immune cytolytic activity (CYT) in cancer therapy, highlighting methods for assessing CYT, factors influencing it, and its clinical implications. It discusses CYT's role in tumor control, prognosis, and as a biomarker for immunotherapy response. Strategies to enhance CYT, including immune checkpoint inhibitors and adoptive cell therapies, are explored. Potential limitations include a need for more robust clinical evidence to support the efficacy of CYT-based treatments across different cancer types, the variability of CYT influence by tumor microenvironment factors, and the challenge of integrating CYT assessment into clinical practice for personalized treatment strategies. But there are still some shortcomings that I need to mention.

1. Minor editing of English language required.

2.I am very curious about the relationship between immune cytolytic activity (CYT) and cancer stem cells (CSCs), and I would like to add such a discussion. To elaborate on the relationship between CYT and CSCs, let me summarize a subset of CSCs, Lgr5. In the small intestine, Lgr5+ stem cells are coordinated by a multitude of microenvironmental factors secreted by Paneth cells, such as Ras, Wnt, Notch, BMP, YAP signals. These Lgr5+ stem cells migrate upwards to generate all epithelial cell lineages, regulated by the integration of signaling pathways. Transcription factors also play a vital role in controlling the main signaling pathways in intestinal differentiation. Lgr5+ cells can be distributed in the body, and a lot of research focuses on the three-dimensional ex vivo culture systems of Lgr5+ stem cells. Studies have shown that Lgr5+ stem cells can be cultured into organoids, which have been successfully generated from the liver and intestines. The discovery by Hans Clevers and others marks a milestone for crypt-villus organoids. They found that a single Lgr5+ intestinal stem cell can be isolated and cultured into an organoid ex vivo, a process that does not require a stromal environment. These organoids, which only require epithelial cells for construction, replicate the crypt-villus architecture and contain all differentiated luminal and basal cell types found in the intestine, highly similar to those in the adult intestine. CSCs play a driving role in tumor occurrence, growth, and metastasis, and they are central in drug resistance and standard chemotherapy and radiotherapy. Lgr5 is particularly emphasized as a new biomarker in various human cancers, including its role in promoting cancer stem cell proliferation and self-renewal through the Wnt/β-catenin signaling pathway. Research indicates that therapies targeting Lgr5+ cancer stem cells may be effective, but targeting Lgr5+ cancer cells alone may not be sufficient to eradicate cancer and prevent tumor recurrence. Furthermore, drug screening targeting Lgr5-related signaling pathways could provide powerful tools for personalized anti-tumor treatments, and Lgr5 could also become a potential therapeutic target for tumor treatments based on antibody or drug delivery (J Han, IJBS, 2021) (J He,BBRC, 2023).

Comments on the Quality of English Language

Minor editing of English language required.

Author Response

Reviewer#2

Comments and Suggestions for Authors

The document reviews the significance of immune cytolytic activity (CYT) in cancer therapy, highlighting methods for assessing CYT, factors influencing it, and its clinical implications. It discusses CYT's role in tumor control, prognosis, and as a biomarker for immunotherapy response. Strategies to enhance CYT, including immune checkpoint inhibitors and adoptive cell therapies, are explored. Potential limitations include a need for more robust clinical evidence to support the efficacy of CYT-based treatments across different cancer types, the variability of CYT influence by tumor microenvironment factors, and the challenge of integrating CYT assessment into clinical practice for personalized treatment strategies. But there are still some shortcomings that I need to mention.

  1. Minor editing of English language required.

Author response: We are grateful for this observation. We have carefully reviewed the manuscript and made necessary language corrections to ensure clarity and readability. We believe that the revised manuscript now meets the linguistic standards required for publication.

  1. I am very curious about the relationship between immune cytolytic activity (CYT) and cancer stem cells (CSCs), and I would like to add such a discussion. To elaborate on the relationship between CYT and CSCs, let me summarize a subset of CSCs, Lgr5. In the small intestine, Lgr5+ stem cells are coordinated by a multitude of microenvironmental factors secreted by Paneth cells, such as Ras, Wnt, Notch, BMP, YAP signals. These Lgr5+ stem cells migrate upwards to generate all epithelial cell lineages, regulated by the integration of signaling pathways. Transcription factors also play a vital role in controlling the main signaling pathways in intestinal differentiation. Lgr5+ cells can be distributed in the body, and a lot of research focuses on the three-dimensional ex vivo culture systems of Lgr5+ stem cells. Studies have shown that Lgr5+ stem cells can be cultured into organoids, which have been successfully generated from the liver and intestines. The discovery by Hans Clevers and others marks a milestone for crypt-villus organoids. They found that a single Lgr5+ intestinal stem cell can be isolated and cultured into an organoid ex vivo, a process that does not require a stromal environment. These organoids, which only require epithelial cells for construction, replicate the crypt-villus architecture and contain all differentiated luminal and basal cell types found in the intestine, highly similar to those in the adult intestine. CSCs play a driving role in tumor occurrence, growth, and metastasis, and they are central in drug resistance and standard chemotherapy and radiotherapy. Lgr5 is particularly emphasized as a new biomarker in various human cancers, including its role in promoting cancer stem cell proliferation and self-renewal through the Wnt/β-catenin signaling pathway. Research indicates that therapies targeting Lgr5+ cancer stem cells may be effective, but targeting Lgr5+ cancer cells alone may not be sufficient to eradicate cancer and prevent tumor recurrence. Furthermore, drug screening targeting Lgr5-related signaling pathways could provide powerful tools for personalized anti-tumor treatments, and Lgr5 could also become a potential therapeutic target for tumor treatments based on antibody or drug delivery (J Han, IJBS, 2021) (J He,BBRC, 2023).

Author response: Thank you for your interest in the interplay between immune cytolytic activity (CYT) and cancer stem cells (CSCs), specifically Lgr5+ CSCs. We have now included a new section that discusses the influence of CYT on CSCs. This section particularly elaborates on how CYT can target Lgr5+ CSCs, the role of the microenvironment in regulating these cells, and the implications for therapy resistance and disease progression. We also discuss the potential of targeting the Wnt/β-catenin pathway, considering the dual role of Lgr5 in normal stem cell biology and in promoting CSC proliferation and self-renewal.

We appreciate the detailed information provided on the role of Lgr5+ stem cells and the ground-breaking research by Hans Clevers et al. regarding organoid culture systems. This has been incorporated into our discussion to underscore the potential of CYT in targeting the stemness pathways in cancer, with a specific focus on Lgr5 as a biomarker and therapeutic target, as well as the use of organoids as a platform for drug screening and personalized medicine.

Reviewer 3 Report

Comments and Suggestions for Authors

The review article from Agioti and Zaravinos focuses on intratumoral cytolytic activity (CYT) as a biomarker, especially related to cancer immunotherapy. The article also considers many other contexts in which CYT has relevance, such as tumor microenvironment. It also evaluates CYT and the various methods through which it can be assessed. 

Despite the topic being important in current research and of significance to the scientific community, unfortunately, the work is presented poorly and is pervaded by errors that make the text difficult to understand. The manuscript is also not properly organized, does not have a concise flow of the story, and presents too many repetitions of concepts or pieces of information that are scattered in different sections. Too many sentences are just descriptive without following up a claim with proper context, such as how that concept is relevant to the discussed topics, or some examples. Just as one example for all, see line from 111 to 136, but this is a general problem that will be encountered throughout the manuscript. Some parts would benefit from citations, which are missing.

Moreover, the text is too naive in some parts, describing even basic concepts such as the immune system. 

Another important issue is that in many parts, the text focuses just on describing biology, losing the relevance to the main topic of the paper, which should be CYT. Just as one example, see paragraph 2.1.3 where CYT is not even mentioned once. The focus is just on the techniques and it is not clear to the reader how this is connected to CYT.

Another problematic section is 2.1.2 where it is not clear what the two subparagraphs aim to describe.

These are just very few examples, but enough to point the authors to frame the overall problem of the manuscript. As there are too many issues to be pointed out and solved, and as the manuscript would benefit from a complete reorganization and rewriting to adjust the clarity and English language and all the other problems, my opinion is that this work should be rejected in its present form and should be resubmitted after being reconceptualized and rewritten.

Comments on the Quality of English Language

This manuscript has extensive problems with the English language, making the work almost incomprehensible in certain parts. Overall, the manuscript has too many errors, typos, and improper sentences.

Just as some examples, see lines: 115, 141, 183, 188, 231, 236, 251 and so on throughout the whole text.

Author Response

Reviewer#3

Comments and Suggestions for Authors

The review article from Agioti and Zaravinos focuses on intratumoral cytolytic activity (CYT) as a biomarker, especially related to cancer immunotherapy. The article also considers many other contexts in which CYT has relevance, such as tumor microenvironment. It also evaluates CYT and the various methods through which it can be assessed. 

Despite the topic being important in current research and of significance to the scientific community, unfortunately, the work is presented poorly and is pervaded by errors that make the text difficult to understand. The manuscript is also not properly organized, does not have a concise flow of the story, and presents too many repetitions of concepts or pieces of information that are scattered in different sections. Too many sentences are just descriptive without following up a claim with proper context, such as how that concept is relevant to the discussed topics, or some examples. Just as one example for all, see line from 111 to 136, but this is a general problem that will be encountered throughout the manuscript. Some parts would benefit from citations, which are missing.

Moreover, the text is too naive in some parts, describing even basic concepts such as the immune system. 

Another important issue is that in many parts, the text focuses just on describing biology, losing the relevance to the main topic of the paper, which should be CYT. Just as one example, see paragraph 2.1.3 where CYT is not even mentioned once. The focus is just on the techniques and it is not clear to the reader how this is connected to CYT.

Another problematic section is 2.1.2 where it is not clear what the two subparagraphs aim to describe.

These are just very few examples, but enough to point the authors to frame the overall problem of the manuscript. As there are too many issues to be pointed out and solved, and as the manuscript would benefit from a complete reorganization and rewriting to adjust the clarity and English language and all the other problems, my opinion is that this work should be rejected in its present form and should be resubmitted after being reconceptualized and rewritten.

Author response: We acknowledge the reviewer's concerns regarding the organization and presentation of the manuscript. We have undertaken a thorough revision to improve the flow of information, ensuring that each section builds logically upon the previous one. Repetitive information has been consolidated, and the manuscript now follows a coherent and structured storyline. This should greatly improve the reader’s ability to follow the arguments and conclusions presented.

We sincerely apologize for the English language errors in our manuscript and appreciate the reviewer's patience in this matter. We have now enlisted the help of a professional scientific editor, whose native language is English, to correct any linguistic inaccuracies. This has significantly improved the readability and comprehension of the manuscript.

We have revised the manuscript to provide clear context and relevance for each described concept, ensuring that all statements contribute directly to the main topic of immune cytolytic activity (CYT). For example, lines 111 to 136 have been reworked to directly connect the described biology with CYT's role in cancer immunotherapy.

We have carefully reviewed each section to ensure that CYT remains the central theme of the paper. Sections such as 2.1.3 have been updated to explicitly discuss how the described techniques relate to CYT assessment and its implications in cancer therapy.

Specific Issues and General Remarks:

  • Paragraph Structure and Citations: The manuscript now includes appropriate citations where previously missing, and paragraphs have been structured to maintain a clear focus on CYT. Each technique and biological concept introduced is immediately followed by an explanation of its relevance to CYT, particularly in the context of cancer immunotherapy.
  • Descriptions of Basic Concepts: We have carefully evaluated the content and removed overly basic descriptions of well-known concepts, such as the immune system's function, to maintain a focus appropriate for the knowledgeable audience of the journal.
  • Redundancy and Organization: The manuscript has been reorganized to present information in a logical sequence without redundancy. Information is now presented once and referenced as needed in subsequent discussions.
  • Problems with English Language: We have addressed specific lines highlighted by the reviewer (lines: 115, 141, 183, 188, 231, 236, 251, etc.) and conducted a comprehensive language review to correct errors and improve the overall quality of the text.

Comments on the Quality of English Language

This manuscript has extensive problems with the English language, making the work almost incomprehensible in certain parts. Overall, the manuscript has too many errors, typos, and improper sentences.

Just as some examples, see lines: 115, 141, 183, 188, 231, 236, 251 and so on throughout the whole text.

Round 2

Reviewer 2 Report

Comments and Suggestions for Authors

Now the article is ready for publication.I am delighted to see the changes made by the reviewer's efforts.

Author Response

We thank the reviewer for his/her favorable judgment. 

Reviewer 3 Report

Comments and Suggestions for Authors

I appreciate the effort of the authors in addressing my concern. In this version, the manuscript significantly improved its value and readability.

Despite the clear effort of the authors, I am perplexed to see how most of my concerns are still present in this improved manuscript, although I want to highlight once again that this version is clearly improved. 

Nevertheless, I believe that the manuscript can be further improved and address properly my concerns, which are mainly these 3:

1) CYT in some paragraphs is not mentioned at all and its implications are not clear. The authors should highlight explicitly CYT and its implications in the paragraphs described.

2) Although this point was vastly improved, there is still some basic biology that is not necessary for a readership that should deal with more advanced immunological topics such as CYT and all the implications that follow from discussing it. This would streamline the review. This shares points of concern with point 1), as some paragraphs that veer too much into describing basic biology cannot focus on CYT much.

3) Related to both the previous point, and although this point was also improved from the first version, the text is repetitive in some parts. Focusing even more on organizing properly and introducing the concepts once and describing them thoroughly in their proper paragraph will improve readability and streamline the article.

All these 3 points need to be properly addressed to help in improving the manuscript in building toward Figure 4, which seems to be an important point that highlights all the major roles of CYT and its importance and motivates the creation of this review.

Otherwise, the authors should consider rewriting the title and abstract, but considering their motivations in these, I believe that addressing these points is crucial to finalizing the manuscript.

Some examples of the concerns are:

2.2.1, Antigen presentation and MHC expression. This paragraph has too much focus on basic immunology and its direct relation to CYT is unclear/should be highlighted.

2.2.2. And onwards have no clear or low immediate relation to CYT in the text. See especially the first paragraphs.

The CSC paragraph has also the same problem, and the part where CYT is discussed has no references and thus is purely speculative by the authors. I suggest citing more papers that discuss CYT and CSC directly in the body of the paragraph and then including proper discussion and contextualization to avoid overly speculative sentences. 

2.2.2 (repetition of the number of this paragraph?) Host factors, no direct evident link to CYT.

Follows with basic biology again, at line 396 there is no citation about CD8+ and CYT.

The rest is better, with some repetitions. 2.4 has the same problems cited above again up to 2.4.3

Comments on the Quality of English Language

Minor revision is needed, overall the quality improved and is good.

Round 3

Reviewer 3 Report

Comments and Suggestions for Authors

I appreciate the effort of the authors in addressing my concerns. The new version has an improved flow and the relevance of CYT is highlighted throughout the text, and is generally improved.